# Anti-Melanogenic Effects of Korean Red Ginseng Oil in an Ultraviolet B-Induced Hairless Mouse Model

**DOI:** 10.3390/molecules25204755

**Published:** 2020-10-16

**Authors:** Evelyn Saba, Seung-Hyung Kim, Yuan Yee Lee, Hyun-Kyoung Kim, Seong-Soo Roh, Yi-Seong Kwak, Chae-Kyu Park, Sung-Dae Kim, Man Hee Rhee

**Affiliations:** 1Department of Veterinary Medicine, College of Veterinary Medicine, Kyungpook National University, Daegu 41566, Republic of Korea; evelyn.saba@uaar.edu.pk (E.S.); yuanyeelee@knu.ac.kr (Y.Y.L.); 2Institute of Traditional Medicine and Bioscience, Daejeon University, Daejeon 34520, Republic of Korea; sksh518@dju.kr; 3Department of Food Science and Engineering, Seowon University, Chungbuk 28674, Republic of Korea; kimhk4@seowon.ac.kr; 4College of Korean Medicine, Daegu Haany University, Daegu 42158, Republic of Korea; ddede@dhu.ac.kr; 5R&D Headquarters, Korean Ginseng Cooperation, Daejeon 34520, Republic of Korea; twostar@kgc.or.kr (Y.-S.K.); ckpark@kgc.co.kr (C.-K.P.); 6Research Center, Dongnam Institute of Radiological and Medical Sciences, Busan 46033, Republic of Korea

**Keywords:** Korean red ginseng oil ointment, melanin, wrinkles, antiaging, collagen

## Abstract

A ‘remedy for all’ natural product widely known in the Korean Peninsula is called *Panax Ginseng* Meyer. Globalization represents a persistent risk to the ozone layer, leading to bountiful amounts of Ultra-Violet B beams (UVB). The variety in human skin hues is ascribed to the characteristic color called Melanin. However, Melanin overproduction due to UVB beams promotes skin staining and tumorigenesis, a process called photo aging, which damages skin quality. To assess the effects of Korean Red Ginseng Oil (KGO) on photo aging, the murine melanoma cell lines B16/F10 were used in vitro and HRM-2 hairless mice exposed to UVB were studied in vivo. Our results revealed that KGO reduced tyrosinase activity and melanin production in B16/F10 cells along with the suppression of upstream factors involved in the melanin production pathway, both transcriptionally and transitionally. In the in vivo studies, KGO suppressed the expression of Matrix Metalloproteinase (MMP) and Interleukins along with a reduction of depth in wrinkle formation and reduced collagen degradation. Moreover, the feed intake and feed efficiency ratio that decreased as a result of UVB exposure was also improved by KGO treatment. In light of our results, we conclude that KGO can have considerable benefits due to its various properties of natural skin enhancement.

## 1. Introduction

Melanin is an intensifying agent that gives our skin its shading. The assortment of skin hues between different humans is credited to melanin creating cells in the body. Melanogenesis is the process that is responsible for melanin formation [1] and α-Melanogenesis is the cycle that is responsible for melanin formation. The α-Melanocyte Stimulating Hormone (α-MSH) is tied to its receptor, i.e., Melanocortin 1 receptor (MC1R), which causes an increased degree of cAMP, which actuates the Microphthalmia associated factor (MITF) through different pathways like the cAMP response element binding protein (CREB), Extracellular Regulating kinase (ERK), and Protein Kinase B (AKT) and causes MITF to degrade. The rate-restricting factor in the melanogenesis process, i.e., the tyrosinase (TYR), becomes affected at this stage. TYR is responsible for increased levels of L-3,4-dihydroxyphenylalanine (L-DOPA) from tyrosine proteins, which shapes melanin. Tyrosinase-related protein 1 (TRP-1) and Tyrosinase-related protein 2 (TRP-2) are further downstream factors that produce melanin through a tyrosinase catalyst and MITF [2,3,4,5]. TYR is thus essential in controlling the measure of melanin creation in skin cells during this cycle.

*Panax ginseng Meyer* is an herb that has been eaten in Eastern Asia over the years and has numerous useful consequences for health. This herb is accessible in various structures, from drinks to tablets. Past and present investigations uncovered the extraordinary action of ginseng in diminishing the frequency of tumors, diabetes, hypertension, hyperlipidemia, irritation, and numerous psychological irregularities [6,7,8].

Korean Red Ginseng Oil (KGO) is a novel herbal commodity prepared from ginseng. Past literature has discussed the anti-inflammatory [9], anti-neuro-apoptotic [10], and anti-hepato-lipidemic effects [11] of KGO. Also, other studies have discussed the safety of KGO for oral use [12], but no research to date has identified the anti-melanogenic effects of KGO on melanin formation and the way it affects skin whitening or its anti-aging properties.

We thus sought to explore the hindrance of TYR and impediments in the improvement of melanin by KGO in vitro in the B16/F10 melanoma cell line by examining the pathway engaged with this process. Our results demonstrated that KGO significantly restrains TYR action and induces a diminished amount of melanin through a system of MITF debasement. Our examination on the Hairless mouse (HRM-2) model uncovered the diminished creation of melanin in HRM-2 mice alongside its antiwrinkle and antiaging properties. Therefore, from the in vivo and in vitro studies, it can be concluded that KGO can be used as a skin whitening and antiaging agent in the cosmetic industry.

## 2. Results

### 2.1. Inhibition of Tyrosinase Activity in Cell-Free System and Melanin Content by Korean Red Ginseng Oil (KGO)

TYR is a rate restricting compound in melanogenesis [13]. Therefore, to assess whether KGO has the capacity to suppress the creation of this chemical, we performed a cell mushroom tyrosinase examination. As shown in Figure 1A, KGO hindered TYR creation. Additionally, we treated the melanoma cell line B16/F10 with KGO and invigorated it with α-MSH to study the melanin emissions. We found that the cell line was productively smothered by KGO, as shown in Figure 1B. In both of these analyses, we used kojic acid as our positive control, as kojic acid is a well-known brightening agent [14].

### 2.2. Impacts of KGO on the mRNA and Protein Articulations of MITF and Tyrosinase-Related Proteins

The above outcomes highlight the impacts of KGO on TYR and melanin substances. Nonetheless, for total comprehension of the mechanism that underlies melanin’s restraint by KGO, we assessed the transcriptional and translational articulation levels of the components in the melanogenesis pathway. These components are TRP-1, TRP-2, TYR, and MITF. As can be found in Figure 2A,B, all the previously mentioned components of the melanogenesis pathway were clearly restrained by KGO. Consequently, we confirmed that KGO is a great brightening specialist at the in vitro level of investigation.

### 2.3. Impact of KGO on the Body Weight of Mice and Food Consumption

Introduction to UVB may induce a loss of appetite. Hence, we assessed the dietary intake and dietary productivity of HRM-2 mice exposed to UVB for 5 weeks and given sunblock with KGO salve applied on the skin. As can be found in Figure 3A, the body weights of the control mice in the UVB group decreased, yet the mice in the positive control with sunblock application and KGO salves demonstrated an increase in body weight when contrasted with the control UVB mice. Likewise, food intake was discovered to be expanded for the positive control and KGO salve groups, as shown in Figure 3B,C. This demonstrates that although UVB may cause stress in mice, KGO salve treatment prevents weight loss.

### 2.4. Examination of Wrinkle-Related Genes in UVB-Induced Skin Damage Model

To gauge the impact of KGO salve treatment on skin-wrinkle arrangement, HRM-2 mice were illuminated with UVB to incite wrinkles. At that point, over the following 5 weeks, skin tissues were separated, and wrinkle-related genes (i.e., IL-1β, MMP-2, and MMP-9) were investigated. It can be seen from Figure 4A,B that the protein levels of MMP-2, as investigated by ELISA and mRNA expressions via qRT-PCR, were fundamentally diminished in the positive benchmark group and KGO salves compared to the control UVB-irradiated mice. Additionally, the mRNA expressions of both the MMP-9 and IL-1β levels were altogether diminished in the 1% KGO-balm-treated mice compared to the control UVB group through qRT-PCR, as shown in Figure 4C,D.

### 2.5. Topical Application of KGO on Melanogenesis in Skin Damage Induced by UVB Irradiation

To confirm the brightening adequacy of KGO salves, the development of melanin on the first, third, and fifth week after UVB light exposure was analyzed by isolating the dorsal skin of HRM-2 mice into the left (untreated) and right (UVB and sample treated areas). As shown in Figure 5A, during the 1st week of the introduction of the mice to UVB with the treatment of sunblock and KGO salves, there was a notable decline in melanin by both KGO balms. At that point, by the 3rd week, there was a large decrease in melanin production in the sunblock-treated groups and 1% KGO ointment group compared with the control UVB group, as indicated in Figure 5B. At that point during the last day of the fifth week (as shown in Figure 5C), the positive control (sunblock)-treated group and both KGO balm-treated groups indicated a critical decrease in melanin creation. These outcomes demonstrate that KGO balm can be utilized as a skin brightening agent.

### 2.6. KGO Prevents Wrinkle Formation and Retains Collagen Content and the Epidermal Parameter in HRM-2 Mice

HRM-2 mice were exposed to UVB to instigate the development of skin damage. Skin wrinkle development was estimated by utilizing a 3D analyzer, as mentioned in the materials and methods, on weeks 3, 4, and 5. As observed in Figure 6A–D, the sunblock-treated group and both KGO balm- treated groups experienced a noteworthy decline in wrinkle development and intensity. Intense radiation can cause an increase in epidermal thickness, making the skin become thicker and unpleasant. Extended exposure to UVB can corrupt the MMPs, causing a decrease in collagen, which is fundamental to great skin health [15,16]. In Figure 7A, the epidermal thickness produced by UVB was attenuated in the sunblock-treated group and both the KGO salve-treated groups. In Figure 7B, the epidermal thickness, as indicated by H&E staining, was diminished in the sunblock-treated group and both KGO-treated groups. M-T staining was performed to visualize the lattice portions in the skin of HRM-2 mice. As shown in Figure 7C, the intensity of the M-T stain decreased in the UVB group due to debasement in the collagen strands. However, for the mice treated with sunblock and both KGO salves, the power of staining was drastically expanded compared to the UVB control groups. These outcomes show that KGO treatments can be utilized for brightening and antiaging results.

## 3. Discussion

Melanocytes are the particular cells that deliver melanin and are found in the epidermal basal layer. Melanocytes are associated with the transportation of melanin to neighboring keratinocytes and structuring the layer of skin pigmentation. Melanin is fundamentally a color that is dual-purpose. Mainly, it provides skin pigmentation and secondly, it guards our skin from destructive Ultraviolet (UV) radiation by reducing the amount of reactive oxygen species [17]. When skin is exposed to UV, the α-MSH factor that actuates melanin fabrication from melanocytes is activated. MITF, the main controlling factor of TYR, TRP-1, and TRP-2, is the principle controller of this whole instrument of melanin creation. These components are involved in melanin development and give our skin its color [18].

Skin maturation involves two mechanisms. One is inborn or endogenous maturing, which is an unavoidable maturation measure experienced by all humans. This type of maturing has similarly gentle side effects, including scarcely discernible differences, unpleasant skin, and decreased elasticity. The second is photo aging or exogenous maturing, which alludes to the maturation cycle known for skin that is exposed to daylight. In this situation, the dependent factor is UVB in sunlight. This type of maturation can be avoided by the prompt use of sun-safe cream and by lessening exposure time to the sun. The clinical indications of exogenous maturation are harmful, producing, for example, harsh and dry skin with diminished flexibility, notable wrinkle arrangement, and significant skin crumbling [19,20].

Likewise, UVB makes skin rapidly engage in pigmentation (e.g., sunlight-based lentigo). Moreover, when the skin is exposed to sunlight, the extracellular network MMPs expand, leading to the debasement of lattice proteins and, among them, collagen. Since collagen is known universally for its versatility in the skin and its maturation, the decay of collagen facilitates the improvement of early winkles and decreases skin elasticity. Increased MMP by UVB corrupts collagen and other base proteins. Wounds are attained by the skin from the beams of the sun, and to recuperate these wounds, new collagen is produced. When the injury healing does not proceed smoothly, the skin will appear more mature with wrinkles [21,22].

In this area, a wide range of natural and solution-based medicines are available, such as sunblock creams and moisturizers for evading the harms from UVB due to daylight exposure while moving under the sun. Physical sun protection like the use of caps and tops that can protect against daylight exposure can also be incorporated to reduce the harm of UVB. Prostaglandin E2, prostaglandin F2α, adrenocorticotropic hormone (ACTH), and nitric oxide are also notable melanogenesis controllers.

However, the impact of cytokines on the cycle of melanogenesis is somewhat confounding. Like the Interleukin (IL) family, IL-1α/1β and the granulocyte-macrophage-state animating element (GM-CSF) are engaged with melanogenesis incitement. However IL-6, TGF-β1, and TNF-α repress the advancement of melanin [23,24]. In our findings, IL-1β levels were expanded in the control UVB populace but were more drastically diminished under treatment with KGO salves. In the foreseeable future, we expect that KGO salves and KGO will have powerful anti-melanogenic and brightening impacts. KGO might likewise be viewed as a suitable restorative agent. However, more studies should be conducted to optimize doses that are safe and potent for human use.

In conclusion, our study proposes the use of KGO as a potent anti-melanogenic agent that can be implemented in skincare routines. The anti-pigmentation or whitening effects of KGO can also be utilized to reduce a dull skin appearance. Oils are also often used in skincare regimens to lock in moisture and prevent dehydration of the skin. Hence, KGO is a suitable candidate for skincare development and also as a therapeutic agent for skin diseases like hyperpigmentation.

## 4. Materials and Methods

### 4.1. Chemicals and Reagents

Dulbecco’s modified Eagle’s medium (DMEM) and fetal bovine serum (FBS) were purchased from WelGene Co. (Daegu, Republic of Korea), streptomycin and penicillin were obtained from Lonza (Walkersville, MD, USA); the TRIZOL^®^ reagent was obtained from Invitrogen (Carlsbad, CA, USA); and oligodT (Bioneer oligo synthesis) and the primers for MITF, TYR, TRP-1, TRP-2, and β-actin were attained from Bioneer (Daejeon, Republic of Korea). 3-(4,5-dimethylthiazol-2-yl)-2,5-diphenyltetrazoliumbromide (MTT), tyrosinase from mushrooms, and L-3,4-dihydroxyphenylalanine (L-DOPA) were procured from Sigma-Aldrich (St. Louis, MO, USA). Antibodies for MITF, TRP-1, TRP-2, and TYR were obtained from Santa Cruz (Santa Cruz Biotechnology Inc., Dallas, TX, USA). All other reagents were of reagent grade.

### 4.2. Sample Preparation and Analysis

Dried ginseng powder was loaded into a pilot-scale supercritical CO_2_ fluid extraction system. The extraction system was set at 6500 psi (relative to 450 bar) at a temperature of 65 °C. RGO was collected in an amber vial and stored at −20 °C. Fatty acid analysis of red ginseng oil was carried out using the method of Min Ji Bak et al. [12] with a SP-2560 column (100 m × 0.25 mm × 0.2 μm; Agilent Technologies, Santa Clara, CA, USA) in a Gas chromatography FID (Flame Ionization Detector) (6890N; Agilent Technologies, Santa Clara, CA, USA). Phytosterol analysis of the red ginseng oil was performed using a DB-1 column (30 m × 0.32 mm × 0.25 μm; Agilent Technolgies, Santa Clara, CA, USA) in the same Gas chromatography FID (Flame Ionization Detector). Compounds of fatty acids and phytosterols in the red ginseng oil were determined by comparing the standard materials matched with both retention time and the area, as reported in a previous study [25].

### 4.3. Cell Line

Murine melanoma cell lines B16/F10, originating from the American Type culture collection (ATCC, Manassas, VA, USA), were cultured in Dulbecco’s Modified Eagle’s Medium (DMEM) supplemented with 8% Fetal Bovine Serum (FBS) (WelGene Co., Daejeon, Republic of Korea) and 100 IU/mL penicillin and 100 μg/mL streptomycin sulfate (Lonza, MD, USA). The cells were cultured in a humidified incubator supplied with 5% CO_2_ at 37 °C.

### 4.4. Cell-free Tyrosinase Inhibition Assay

The assay was performed with slight modifications as previously described [26]. Briefly, 70 μL of RGO or kojic acid (20 μM) was added to 20 μL of mushroom tyrosinase (10 μg/mL), followed by the addition of 10 μL of L-DOPA (10 mM). Plates were incubated for 10 min and read at 475 nm using a plate reader (Versamax; Molecular Devices, LLC, San Jose, CA, USA).

### 4.5. Melanin Inhibition Assay

The melanin inhibition assay was slightly modified from that previously described [27]. Briefly, B16/F10 cells were seeded and incubated for 5 days in 6-well culture plates, followed by KRG and α-MSH treatments. The cells were left to further incubate for 3 more days and were then harvested and centrifuged at 10,000 rpm for 10 min, and the cell pellets were dissolved in NaOH at a concentration of 2 mol/L and left in a water bath of 60 °C for 15 min. The subsequent mixture was analyzed at an absorbance of 450 nm using a plate reader (Versamax; Molecular Devices, LLC, San Jose, CA, USA).

### 4.6. Animal Experiment Regime and Grouping

Six-week-old male HRM-2 melanin-possessing hairless mice were obtained from Central Lab Animal Inc. (Seoul, Republic of Korea) and housed in a controlled room (23 ± 1 °C, 55 ± 5% humidity, 12-h light/dark cycle) with *ad libitum* access to water and feed. All animal experiments were strictly carried out according to the Institutional Animal Care and Use Committee of Daejeon University (Daejeon, Republic of Korea) (Permission number: DJUARB2017-033). After 1 week of acclimation, the mice were randomly divided into 5 groups (5 animals per group). The first group was a normal group with no treatment at all. The second group was the control UVB-treated group, and the third group was a positive control group that received 0.01% sunblock with UVB. The fourth group received KGO (1%) ointment with UVB, and the fifth group received KGO (0.5%) ointment with UVB. All efforts were made to reduce the number of animals and minimize harm to the animals.

### 4.7. UVB Irradiation in Mice and induction of Photo Aging

HRM-2 mice were irradiated on the dorsal skin with a UVB lamp (15 W, maximum wavelength 312 nm; UV intensity 100 µWcm^−2^, IedaBoeki Co., Tokyo, Japan). The positive control (0.01% sunblock) and KGO ointments were applied daily, 5–10 min before exposure of the mice to UVB radiation. HRM-2 mice were exposed to 100 mJ/cm^2^ UVB radiation (1 minimal erythematic dose = 100 mJ/cm^2^) daily for the first week; then, the UVB radiation was increased to 200 mJ/cm^2^ from 2 to 5 weeks, and the mice were monitored 3 times each week. Dietary intake and body weights were measured at regular intervals weekly up to 12 weeks.

### 4.8. Effects of KGO on Melanin Production in HRM-2 Mice

The protocol for this experiment was conducted according to our previously reported study [27]. Briefly, pigmentation of the mice was assessed using a Nikon D70 camera (Nikon, Tokyo, Japan) that was fixed using a stand to maintain a distance of 20 cm from the mice. Images were captured and analyzed via software (Biorad, Hercules, CA, USA), and images of the pigmented left side and unpigmented right side were compared.

### 4.9. Skin Wrinkle Assessment

UVB-induced skin aging was measured by observing wrinkle formation. To assess the formation of wrinkles, we used the protocol previously described [27]. Briefly, wrinkles were evaluated on weeks 3, 4, and 5 with a DETAX System II (MIXPAC) and a Double-Stick Disc (3M Healthcare, St. Paul, MN, USA). The disc was attached to the skin of the mice and left for 2–3 min. Wrinkles were scored and photographed with a USB Digital Microscope at 400X (CE FOROHS, China).

### 4.10. Enzyme-Linked Immunosorbent Assay (ELISA)

UVB-induced wrinkle-related quality, MMP-2, was investigated in the skin of all treated groups. The ELISA investigation was carried out as indicated by the manufacturer (MMP-2 ELISA unit) (R&D Systems, Minneapolis, MN, USA). According to manufacturer’s instructions, the sensitivity for the MMP-2 ELISA kit was 0.014–0.082 ng/mL. This kit is specifically used for Recombinant MMP-2 and natural human, mouse, rat, porcine, and canine active, pro-, and TIMP complexed MMP-2. The detection range for this kit was 0.5–32 ng/mL.

### 4.11. Histological Observation of Skin

To assess histological lesions, the skin tissues were extracted and immediately fixed in 10% neutral buffered formalin for 48 h. Then, Haematoxylin and Eosin staining (H&E) was performed according to [28] to determine epidermal thickness. To visualize collagen, Masson’s Trichome (M-T) staining was subsequently performed on the collagen according to previously reported protocols [29].

### 4.12. RNA Extraction and qRT-PCR

After treatment with KGO and stimulation with α-MSH and KGO ointments, total RNA was extracted from the B16/F10 cells and from the UVB-irradiated mouse skin with TRIZOL^®^ according to the manufacturer’s instructions. Subsequent steps were followed according to our previously reported study [30]. The primer sequences are listed in Table 1.

### 4.13. Western Blot Analysis

B16/F10 cells were treated with KGO in the presence of α-MSH (10µM). Protein was extracted from the cells and the mouse skin of the UVB-irradiated mice according to the manufacturer’s instructions for the PRO-PREP^®^ lysis buffer (iNtRON Biotechnology, Republic of Korea). The preceding steps were performed according to our previously reported study [30]. Briefly, the protein was separated on 10% SDS-PAGE and transferred onto PVDF membranes, followed by blocking with skim milk. Membranes were washed and incubated with a primary antibody at a dilution of 1:1000 overnight at 4 °C on a roller. The next day, the membranes were washed and incubated with a secondary antibody (1:3000 dilution) for 75 min before developing enhanced chemiluminescence in a gel developer (General Electrics, Boston, MA, USA).

### 4.14. Statistical Analysis

Data are presented as the mean ± SEM. A one-way ANOVA, Dunnett’s test, and unpaired student’s T test were applied for statistical evaluation of the data. Statistical analyses with **** *p* <* 0.001, ** *p* < 0.05, and * *p* < 0.01 were considered significant compared to the UVB-control.

## Figures and Tables

**Figure 1 molecules-25-04755-f001:**
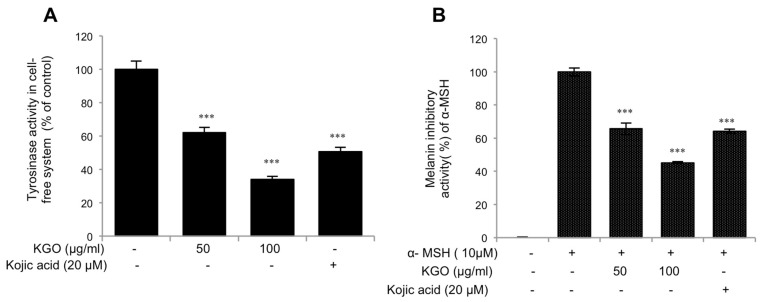
KGO (Korean Red Ginseng Oil) inhibits mushroom tyrosinase and the production of melanin. (**A**) KGO inhibited mushroom tyrosinase activity in a cell free system. (**B**) The content of melanin in crude lysates was suppressed by KGO in B16/F10 cells stimulated with α-MSH. Values in the bar graph are the mean ± SEM of at least 4 independent experiments. ****p* < 0.001 compared to the untreated group.

**Figure 2 molecules-25-04755-f002:**
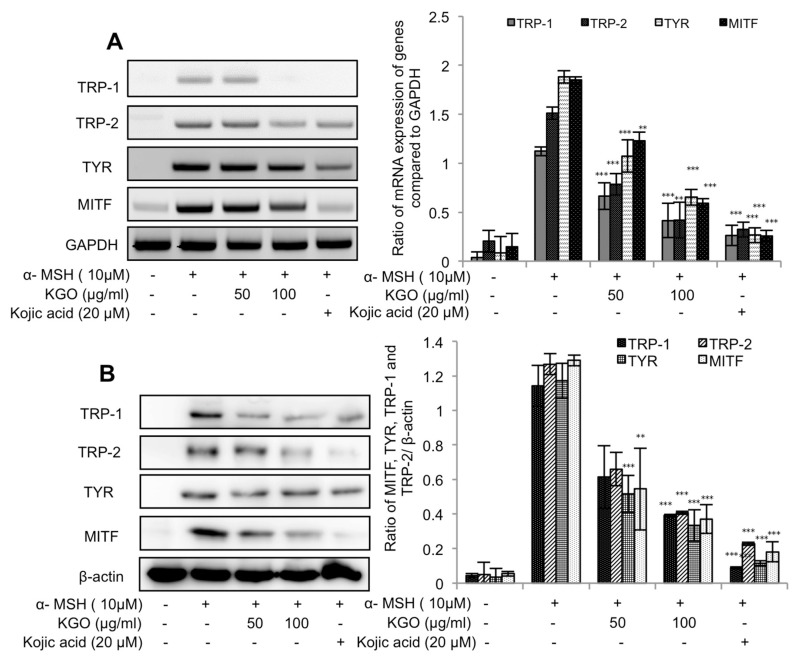
Suppressed expression of genes in the MITF (microphthalmia associated factor) pathway by KGO in B16/F10 cells. Cells were seeded in 6-well plates, treated with the indicated concentrations of KGO, and then stimulated with α-MSH (10 µM). Total RNA and proteins were extracted, and the TRP-1, TRP-2, TYR, (tyrosinase) and MITF expression levels were assessed by qRT-PCR (**A**) and Western blotting (**B**). GAPDH (glyceraldehyde 3-phosphate dehydrogenase) was taken as the internal control for assessing transcriptional expression, and β-actin was taken as the translational control, and all values were compared against them. *** *p* < 0.001, ** *p* < 0.05, and * *p* < 0.01 were considered as statistically significant against the α-MSH-treated group only.

**Figure 3 molecules-25-04755-f003:**
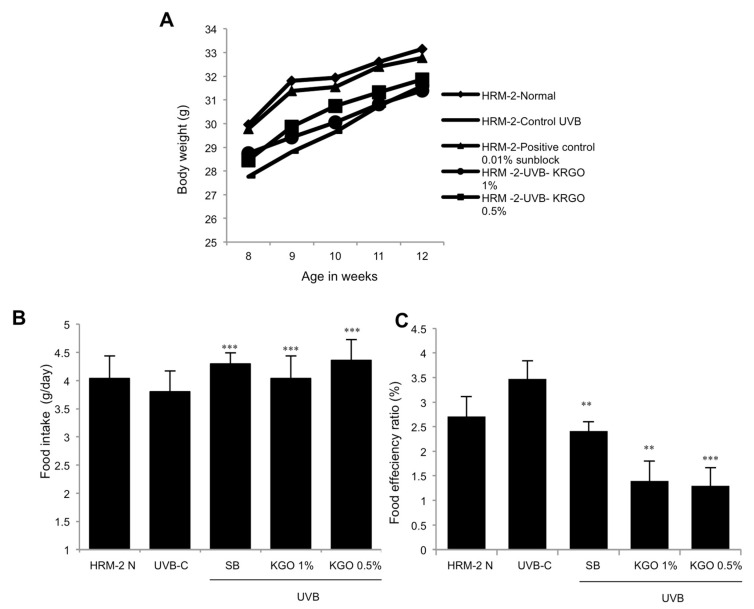
The effects of KGO ointments on body weight change and food intake in mice. To measure body weight and food intake, HRM-2 (hairless mice) mice were exposed to UVB (Ultra-Violet B beams) for 5 weeks with KGO ointments and positive control treatments on the skin. Food intake and body weight were assessed daily for 5 weeks. (**A**) KGO ointments substantially increased body weight compared to the UVB control. (**B**) Daily food intake was significantly increased for the positive control and both the KGO ointment groups. (**C**) Feed efficiency ratio (FER). Values are expressed as the mean ± SEM from three -independent experiments. **** p* < 0.001 and *** p* < 0.05 when compared with the UVB control. The normal mice are abbreviated as HRM-2 N; the UVB control is abbreviated as UVB-C; sunblock is abbreviated as SB.

**Figure 4 molecules-25-04755-f004:**
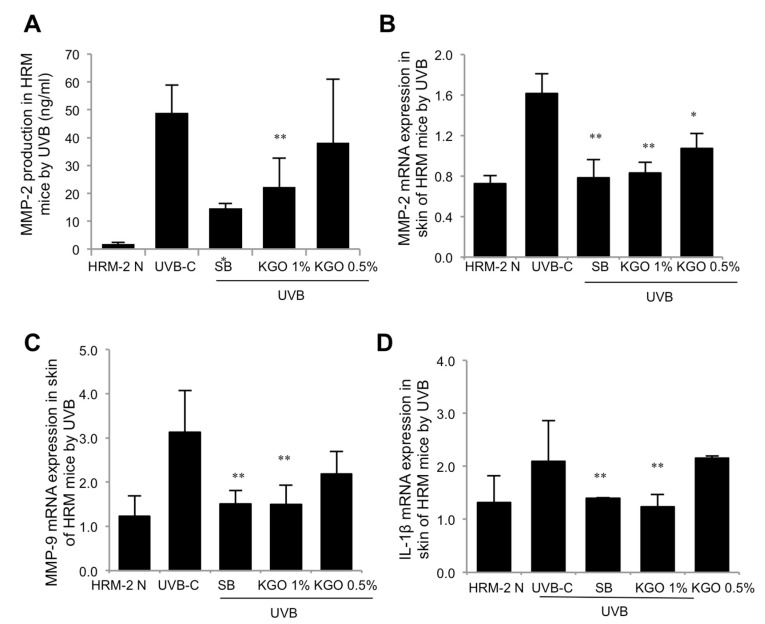
Investigation of wrinkle-related gene expressions in UVB-irradiated mice. HRM-2 mice were exposed to UVB for wrinkle formation for 5 weeks along with the positive control and the KGO ointment treatments. Five weeks later, skin tissues were isolated, and MMP-2, MMP-9, and IL-1β gene expression was analysed. (**A**) MMP-2 production was significantly suppressed by 1% KGO ointment, as determined by ELISA. (**B**) The mRNA expression of MMP-2 was significantly decreased by both KGO ointment groups, as determined by qRT-PCR. (**C**) MMP-9 and (**D**) IL-1β mRNA expression was significantly decreased by 1% KGO ointment, as determined by qRT-PCR. Values in the bar graphs are ± SEM from three-independent experiments ** *p* < 0.05, and * *p* < 0.01 when compared with the UVB control. The normal mice are abbreviated as HRM-2 N; the UVB control is abbreviated as UVB-C; sunblock is abbreviated as SB.

**Figure 5 molecules-25-04755-f005:**
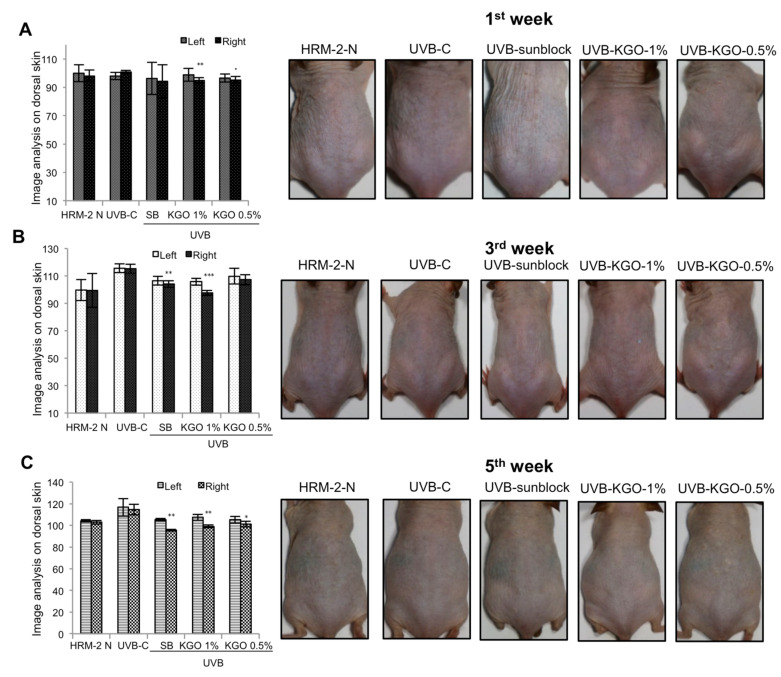
Effects of KGO ointment on Melanogenesis in Skin-Damage-Induced UVB Irradiation in Mice. To elucidate the whitening efficacy of KGO, the formation of melanin on 1st, 3rd, and 5th weeks of UVB irradiation was analysed by dividing the dorsal skin of HRM-2 mice into the left (untreated) and right (treated) sides. Images were taken on the (**A**) 1st week, (**B**) 3rd week, and (**C**) 5th week. Values in the bar graphs are ± SEM from three independent experiments. *** *p* < 0.001, ** *p* < 0.05, and * *p* < 0.01 when compared with the UVB control. The normal mice are abbreviated as HRM-2 N; the UVB control is abbreviated as UVB-C; sunblock is abbreviated as SB.

**Figure 6 molecules-25-04755-f006:**
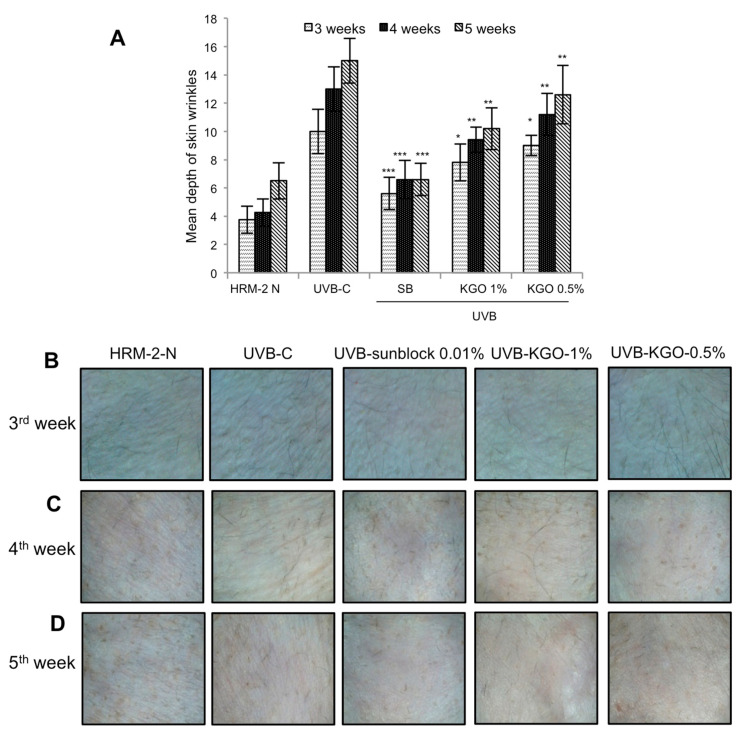
KGO ointment on wrinkle formation in HRM-2 mice. To measure the effect of improving the wrinkle depth with KGO ointments, HRM-2 mice were exposed to UVB to induce photo aging. The skin wrinkle depth was analysed, and images were taken (**A**). Changes in the depth of the wrinkles on the 3rd, 4th, and 5th weeks of KGO ointment treatment were observed. Values in the bar graphs are ± SEM from the experiments repeated in triplicate. *** *p* < 0.001, ** *p* < 0.05, and * *p* < 0.01 when compared with the UVB control. (**B**) Wrinkle formation on the 3rd week of KGO ointment treatment. (**C**) Wrinkles formation on the 4th week of KGO treatment. (**D**) Wrinkle formation on the 5th week of KGO ointment treatment. The normal mice are abbreviated as HRM-2 N; the UVB control is abbreviated as UVB-C; sunblock is abbreviated as SB.

**Figure 7 molecules-25-04755-f007:**
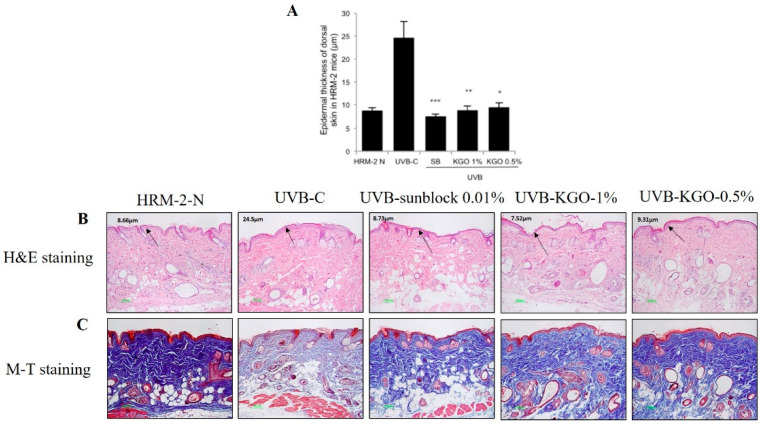
Effects of KGO ointment on epithelial thickness and changes in collagen fibre in HRM-2 mice. After 5 weeks, the epithelial thickness of the skin was observed after staining with H&E (**A**,**B**). A significant reduction was found in the epidermal thickness of the KGO ointment-treated groups. Values in the bar graphs are ± SEM from three independent experiments. *** *p* < 0.001, ** *p* < 0.05, and * *p* < 0.01 when compared with the UVB control. (**C**) The intensity of M-T staining was decreased in the UVB-control group compared to the normal group, suggesting that collagen fibre degradation progressed, and wrinkle formation accelerated. However, the amount of collagen fibres in the KGO ointments (1 and 0.5%) and positive control group increased, indicating that KGO ointments reduced the amount of collagen degradation. The normal mice are abbreviated as HRM-2 N; the UVB control is abbreviated as UVB-C; sunblock is abbreviated as SB.

**Table 1 molecules-25-04755-t001:** Oligonucleotide sequences of primers/probes used for qRT-PCR.

Genes	Primer/Probe	Oligonucleotide Sequence (5′-3′)	Accession Number
MITF	Forward Reverse	5′-CATGCAGTCCGAATCGGGAA-3′ 5′-ACTGTTCTCCTCCCAGGGTA-3′	XM_036165908.1
TYR	Forward Reverse	5′-AATTCTGGGCTCAGAGATGTTT-3′ 5′-ACTTCTTGCCTGGCCTACAC-3′	JN956476.1
TRP-1	Forward Reverse	5′-GTCCAATAGGTGCGTTTTCC-3′ 5′-ACCCATTTGTCTCCCAATGA-3′	XM_006537781.2
TRP-2	Forward Reverse	5′-TAGCTGCTTCTGGTGGCAAG-3′ 5′-ATAAGTACACACACGGGGCG-3′	X85126.1
MMP-2	Forward Reverse	5′-CAGGGAATGAGTACTGGGTCTATT-3′ 5′-ACTCCAGTTAAAGGCAGCATCTAC-3	NM_008610.3
MMP-9	Forward Reverse	5′-AATCTCTTCTAGAGACTGGGAAGGAG-3 5′-AGCTGATTGACTAAAGTAGCTGGA-3′	XM_006498861.3
IL-1β	FAM	5′-CTGTGTAATGAAAGACGGCACACCCACC-3′	XM_006498795.5

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
