# Peer review of "Anti-Melanogenic Effects of Korean Red Ginseng Oil in an Ultraviolet B-Induced Hairless Mouse Model"

_molecules, 2020, doi:10.3390/molecules25204755_

Round 1

Reviewer 1 Report

Thank you for submitting the manuscript “Anti-melanogenic effects of Korean Red Ginseng Oil in Ultraviolet B-induced Hairless mouse model” to Molecules. The manuscript is very interesting because the authors showed that KGO reduced tyrosinase activity and melanin production in B16/F10 cells along with suppression of upstream factors found to be involved in melanin production pathway. Therefore, in my opinion minor revision needs to be carried out in the text.

  • Check all the text regarding the space between the numbers and the units
  • In item 4.8, describe better how the photos were obtained, for example, what is the distance between the object of the figure and the camera?
  • For publication, figures need to be improved in terms of resolution

Author Response

Reviewer 1

Thank you for submitting the manuscript “Anti-melanogenic effects of Korean Red Ginseng Oil in Ultraviolet B-induced Hairless mouse model” to Molecules. The manuscript is very interesting because the authors showed that KGO reduced tyrosinase activity and melanin production in B16/F10 cells along with suppression of upstream factors found to be involved in melanin production pathway. Therefore, in my opinion minor revision needs to be carried out in the text.

  1. Check all the text regarding the space between the numbers and the units

Answer: Respected Sir/Madam! Thank you for your useful comment. We have checked the text regarding space between numbers and units respectively. We have changed the text color to red for changes made in Section 4.2, 4.3, 4.4, 4.5, 4.6, 4.7, 4.9 and 4.13 on pages 8-12 respectively.

  1. In item 4.8, describe better how the photos were obtained, for example, what is the distance between the object of the figure and the camera?

Answer: Respected Sir/Madam, thank you for your suggestion. The picture of mice were taken by a Nikon D70 camera and a distance of 20cm was maintained using a stand between the object and camera lens. We have added this statement in line 308, shown in red. We thank you for pointing this out to us.

  1. For publication, figures need to be improved in terms of resolution

Answer: Respected Sir/Madam, we thank you for your suggestion. We have improved the resolution and increased the size of the figures.

Reviewer 2 Report

This manuscript investigates the effect of Korean Red Ginseng Oil (KGO) on melanogenosis in cell lines and UV-induced hairless mouse. Authors concluded that KGO might have some benefits for various properties of natural skin enhancement.

Major Comments:

  1. Abstract: more results should be addressed. It is hard to understand the finding of this manuscript.

  1. The grammar for this manuscript is OK! Whole writing, however, is redundant, and difficult to read. An English editing is essential for this manuscript.

Minor comments:

  1. Page 9, ELISA for MMP-2, please show the sensitivity, detection range, and specificity from the manufacturer’s direction.

  1. page 10, Acknowledgments: the names of colleagues and students should be listed.

Author Response

Reviewer 2

This manuscript investigates the effect of Korean Red Ginseng Oil (KGO) on melanogenosis in cell lines and UV-induced hairless mouse. Authors concluded that KGO might have some benefits for various properties of natural skin enhancement.

Major Comments:

  1. Abstract: more results should be addressed. It is hard to understand the finding of this manuscript.

Answer: Respected Sir/Madam, we have added some more result details in the whole abstract section and changed the color to red.

  1. The grammar for this manuscript is OK! Whole writing, however, is redundant, and difficult to read. An English editing is essential for this manuscript

Answer: Respected Sir/Madam, we have rechecked the manuscript for grammatical errors. We have also attached the certificate for English editing for this manuscript as below. We would like to thank you again for providing us useful suggestions to improve our work.

Minor comments:

  1. Page 9, ELISA for MMP-2, please show the sensitivity, detection range, and specificity from the manufacturer’s direction.

Answer: Respected Sir/Madam, we thank you for your useful comment. We have added the desired details on page No. 9, Section 4.10, lines 320-323 respectively.

  1. Page 10, Acknowledgments: the names of colleagues and students should be listed.

Answer: Respected Sir/Madam, thank you for your concern, we have added the name of the colleagues and students that has provided support in this study in the acknowledgement section, shown in red.

Round 2

Reviewer 2 Report

No more comment.